# RETHINKING AND BENCHMARKING LARGE LANGUAGE MODELS FOR GRAPH REASONING

## ABSTRACT

Large Language Models (LLMs) for Graph Reasoning have been extensively studied over the past two years, involving enabling LLMs to understand graph structures and reason on graphs to solve various graph problems, with graph algorithm problems being the most prevalent. Recent studies underscore the potential of LLMs in handling graph reasoning tasks, but their performance is underwhelming. In this work, we point out issues with existing methods and benchmarks, and rethink the direction that LLMs for graph reasoning should strive toward. We find that base models, e.g., GPT-4o-mini, are largely underestimated due to improper reasoning focus. Base models with reasoning focus redirected from replicating graph algorithms to designing them can easily solve most graph reasoning tasks in existing benchmarks. To truly evaluate the graph reasoning capabilities of LLMs, we construct a more challenging GraphAlgorithm benchmark, comprising 239 different graph problems and 3,041 test instances collected from 4 competition platforms. Finally, we introduce a simple and strong baseline Simple-**R**easoning-**T**hen-**C**oding (Simple-RTC)—which guides LLMs to design graph algorithms first and then code to address graph reasoning tasks. Simple-RTC achieves near-perfect accuracy on existing benchmarks and significantly outperforms GPT-4o-mini and all prior methods on the GraphAlgorithm benchmark. This strong baseline encourages further advancements in LLMs for Graph Reasoning in the future. Code is available at https://anonymous.4open.science/r/Simple-RTC-B58D.

## 1 INTRODUCTION

Graphs play a crucial role in modeling complex real-world relationships. Many significant applications like drug discovery (Stokes et al., 2020), traffic forecasting (Jiang & Luo, 2022), and financial detection (Motie & Raahemi, 2024) are essentially graph problems. Due to the irregular and complex nature of graphs, it often requires graph experts to conduct specialized analysis for each graph problem. Considering that most people lack knowledge of graphs, researchers have explored using Large Language Models (LLMs) to solve graph problems (Wang et al., 2023; Chen et al., 2023; Zhang et al., 2024c). Typically, graph problems are provided in textual form, and then LLMs analyze the graph and reason to generate solutions in natural language form.

To enhance the capability of LLMs in solving graph problems, various works (Chen et al., 2024; Chai et al., 2023; Fatemi et al., 2024; Zhang et al., 2024a; Li et al., 2024b) have been proposed to learn and address different graph reasoning tasks, ranging from simple node counting tasks to NP-hard problems like the traveling salesman problem (TSP). There are primarily two types of approaches: language-based and code-augmented methods, as shown in Figure 1. Language-based methods rely entirely on language to replicate the process of graph algorithms for reasoning, thus are heavily affected by "*repetitive iterative and backtracking operations*" problems (detailed in Section 3.1). Code-augmented methods depend on graph algorithm APIs and external knowledge bases to solve problems, primarily enhancing the model's ability to invoke various tools rather than utilizing the model's inherent graph reasoning capabilities. Furthermore, we find that guiding base models like GPT-4o-mini to first design graph algorithms for solving problems and then implement them through programming, rather than directly replicating algorithms on the input graph, can significantly enhance the performance across various graph reasoning tasks, as shown in Figure 2.

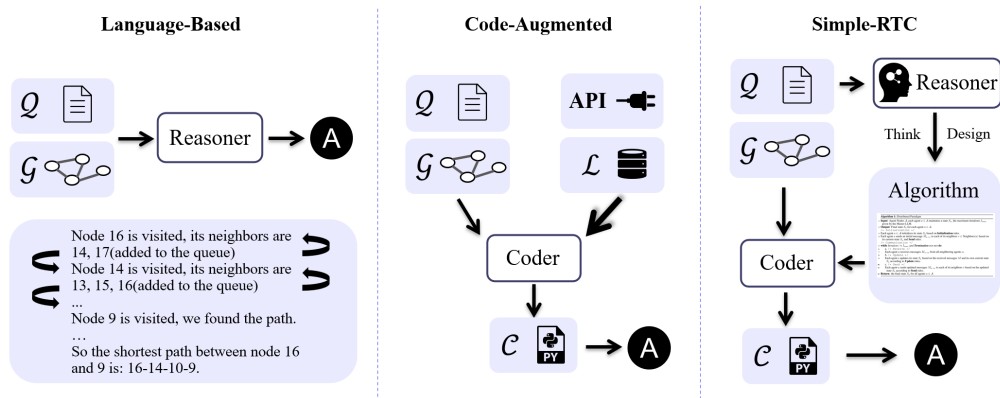

Figure 1: Illustration of Language-Based, Code-Augmented methods and our Simple-RTC.

Moreover, existing benchmarks primarily focus on classical graph algorithm problems (Fatemi et al., 2024; Wang et al., 2023; Luo et al., 2024; Chen et al., 2024; Tang et al., 2024) that can be readily solved using standard graph APIs such as NetworkX (Hagberg & Conway, 2020). Evaluation settings where testing tasks are identical to those in the training set are also problematic, as models may simply memorize fixed patterns rather than develop genuine graph reasoning capabilities. To better evaluate the capabilities of LLMs in solving graph problems, we manually curated problems from online competition platforms, including Codeforces, AtCoder, CodeChef, and Kattis, to construct a new benchmark named **GraphAlgorithm**, featuring challenging problems with realistic descriptions. After filtering and processing, we collected 239 different graph reasoning problems and 3,041 test instances, providing a comprehensive and meaningful evaluation of graph reasoning capabilities.

Finally, inspired by the findings in Figure 2, we design a simple **R**easoning-**T**hen-**C**oding method——Simple-RTC. We solve graph problems by decoupling graph reasoning and coding. On one hand, graph reasoning focuses on analyzing problems and designing graph algorithms instead of reproducing the complete derivation process; on the other hand, graph coding translates the algorithms from reasoning into programs to precisely handle the extensive operations involved.

In summary, our contributions are as follows:

- We revisit LLMs for Graph Reasoning and identify issues preventing progress. Language-based methods are significantly affected by "*repetitive iterative and backtracking operations*" problems; code-augmented methods overlook the critical reasoning in addressing graph reasoning problems.

- We benchmark LLMs for Graph Reasoning with GraphAlgorithm. Problems in GraphAlgorithm can hardly be solved by classical graph algorithms, requiring the model to design graph algorithms, thereby providing a better assessment of the model's graph reasoning capabilities.

- We refine LLMs for Graph Reasoning by designing the Simple-RTC baseline. By guiding LLMs to focus their reasoning on the design of graph algorithms, Simple-RTC improves the performance of the base model GPT-4o-mini by **39%** to **62%** across various benchmarks.

## 2 PRELIMINARIES

### 2.1 GRAPH REASONING PROBLEMS

The Graph Reasoning Problem involves the task of extracting, inferring, or predicting meaningful relationships and properties from graph-structured data. Given a graph composed of nodes and edges, the goal is to reason about the underlying structure, connectivity, or semantic relationships encoded within the graph. This often requires leveraging both the local neighborhood information of nodes and the global topology of the graph. The problem spans a wide range of applications, including classic graph algorithm problems, node classification, link prediction, graph classification, and is fundamental to tasks in domains such as social networks, knowledge graphs, and molecular

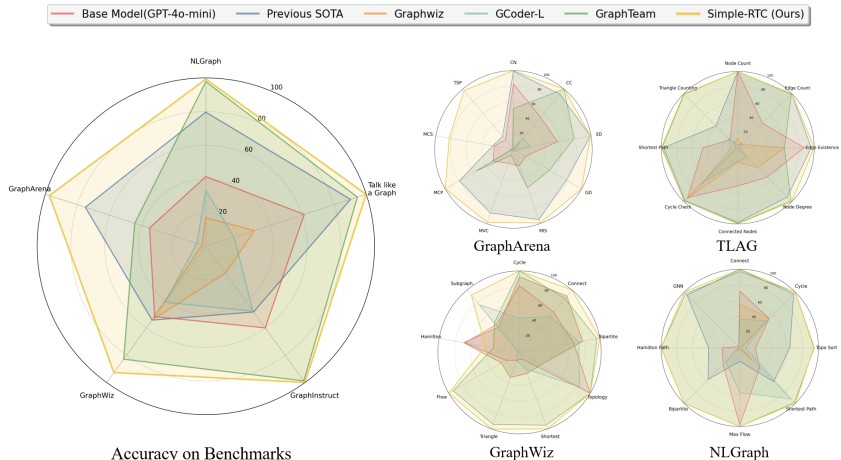

Figure 2: The performance of different methods on benchmarks (left) and different tasks within GraphArena, Talk like a Graph, GraphWiz and NLGraph (right). The red bars are the baseline model, GPT-4o-mini (reasoning to replicate graph algorithms), while the golden bars are our proposed Simple-RTC model (reasoning to design graph algorithms), which also uses GPT-4o-mini as the base model. We can see the shift in reasoning focus has led to a comprehensive and significant improvement in performance.

biology. To comprehensively evaluate the capabilities of LLMs in understanding graph structures and reasoning over graphs, the current works primarily focus on various graph algorithm problems.

## 2.2 LLMs for Graph Reasoning

In general scenarios, when discussing LLMs solving graph reasoning problems, the input is a $(\mathcal{G},\mathcal{Q})$ pair. $\mathcal{G}$ is a graph represented as $\mathcal{G} = (\mathcal{V}, \mathcal{E}, \{s_i\}, \{t_i\})$, where $\mathcal{V}$ is the node set and $\mathcal{E}$, the edge set. For each node $v_i \in \mathcal{V}$, a sequential text node feature $s_i$ is associated; similarly, for each edge $e_i \in \mathcal{E}$, a sequential text edge feature $t_i$ is assigned. The graph $\mathcal{G}$ is described in natural language, typically using edge or adjacency list representation. $\mathcal{Q}$ is a task-specific instruction. Most of the previous methods are language-based, which reason through natural language. Specifically, the model $\mathcal{M}$ processes the $(\mathcal{G},\mathcal{Q})$ pair and outputs the answer $\mathcal{A}$ in textual form:

$$\mathcal{M}(\mathcal{G}, \mathcal{Q}) \to \mathcal{A}.$$

Recently, recognizing the limitations of language-based methods in handling graph reasoning tasks, some works have proposed code-augmented methods. In these approaches, the model $\mathcal{M}$ generates a program $\mathcal{C}$ based on the problem, and the final answer is obtained by executing the program:

$$\mathcal{M}(\mathcal{G}, \mathcal{Q}) \to \mathcal{C}, \mathrm{exec}(\mathcal{C}) \to \mathcal{A}.$$

## 3 Issues with Existing Methods

We analyze existing works categorized into language-based and code-augmented methods. For each method, the analysis will be emphasized on its defect found in the process of solving different graph reasoning problems by using its official code and setting as its original paper.

### 3.1 Language-Based Methods

#### 3.1.1 Prompt Engineering Methods

NLGraph (Wang et al., 2023), LLMtoGraph (Liu & Wu, 2023), GPT4Graph (Guo et al., 2023), LLM4DyG (Zhang et al., 2024c), and Talk like a Graph (Fatemi et al., 2024) are among the early works exploring the use of LLMs for graph reasoning tasks. These works primarily investigate various methods for representing graph structures, such as traditional edge or adjacency list representations,

as well as more sophisticated formats like Graph Modeling Language (GML) (Himsolt, 1997) and Graph Markup Language (GraphML) (Brandes et al., 2013). They explore the impact of different prompting strategies, such as zero-shot, few-shot (et al., 2020), and chain-of-thought (Kojima et al., 2022), on the ability of LLMs to understand graph structures and perform graph reasoning.

A common issue shared by these methods is that graph structures are too complex for LLMs to memorize. As graphs scale up and become denser, they become highly complex, requiring a large number of tokens to describe. However, LLMs tend to perform worse when handling long texts (Liu et al., 2023), with inference time also increasing significantly. To achieve correct reasoning results, LLMs need to replicate the reasoning paths of graph algorithms. However, due to the excessive iterative and backtracking operations in graph algorithms, it becomes challenging to output a complete and correct reasoning path during inference. This issue becomes more severe as graph size increases. Most of the following language-based methods also suffer from the same "*repetitive iterative and backtracking operations*" problem, which leads to low accuracy and poor scalability.

### 3.1.2 FINE-TUNING METHODS

**GraphWiz (Chen et al., 2024)**   GraphWiz first constructs a graph reasoning dataset with explicit reasoning paths and trains its model on this dataset using Supervised Fine-Tuning (SFT) and Direct Preference Optimization (DPO) (Rafailov et al., 2023). The resulting model is capable of outputting explicit reasoning paths to solve graph reasoning problems. However, GraphWiz overfits to the nine graph reasoning tasks it was trained on. When tested on other graph problems, it incorrectly applies the templates from the training problems.

**GraphToken (Perozzi et al., 2024)**   GraphToken uses various Graph Neural Networks (GNNs) (Kipf & Welling, 2017) to encode graph structures into GraphTokens and attempts to align the GraphTokens with the text tokens comprehensible to LLMs. The primary issue with GraphToken is its reliance on task-specific graph encoders that require separate training for different tasks. When a graph encoder trained on one task is applied to others, the model shows a significant performance drop, reducing its practical applicability.

**GITA (Wei et al., 2024)**   Inspired by humans' intuitive understanding of graph structures through visualization, **G**raph to v**I**sual and **T**extual Integr**A**tion (GITA) employs a Visual Large Language Model (VLLM) to interpret graph structures and then perform reasoning at the language level. However, GITA is also limited to handling smaller-scale graphs. As the number of nodes and edges increases, the visual graphs generated by tools like Graphviz (Gansner & North, 2000) and Matplotlib (Tosi, 2009) become excessively complex and difficult for VLLMs to comprehend, thereby offering limited assistance—or even potentially having counterproductive effects on LLMs' reasoning.

### 3.2 CODE-AUGMENTED METHODS

**CodeGraph (Cai et al., 2024)**   CodeGraph employs few-shot sample code prompting to guide LLMs to solve graph problems through code generation, addressing the computation errors that frequently arise during algorithm execution in language-based methods. However, CodeGraph requires manually crafted problem descriptions and sample code for each individual problem, rendering it unable to independently solve unseen graph problems and limiting its generality and practicality.

**GraphTeam (Li et al., 2024b)**   GraphTeam utilizes multi-agent to solve graph problems. Specifically, it constructs a Knowledge Base consisting of documentation for graph-related APIs (e.g., NetworkX) and accumulated experiences from the execution process. During inference, the search agent retrieves relevant API documentation or execution experiences from the knowledge base and provides them as context to the coding agent, enabling it to write code to solve graph problems. However, when applied to algorithmic problems not included in its knowledge base, GraphTeam performs relatively poorly, as shown in Table 2 and Table 3.

**GCoder (Zhang et al., 2024a)**   GCoder enhances the ability of LLMs to utilize graph APIs through SFT and Reinforcement Learning from Compiler Feedback (RLCF). For graph tasks not included in training sets, GCoder retrieves similar code from the code library it builds as additional context to assist in code generation. One issue with GCoder is that it focuses on learning how to invoke various

graph algorithm APIs rather than learning the algorithms themselves, which does not enhance the model's graph reasoning capabilities. When testing GCoder on other graph tasks, its performance is unsatisfying, as shown in Table 1. This might be caused by Multi-Task Finetuning, where the model performs well on the training tasks but poorly on other graph tasks, similar to GraphWiz.

### 3.3 SUMMARY

In summary, language-based methods generally suffer from poor scalability, low accuracy, and incomplete reasoning paths. A recent work (Hu et al., 2024) utilizes multi-agent to simulate the complete graph reasoning process. While this approach can achieve relatively high accuracy, the inference time and cost increase significantly as the number of nodes grows.

Code-augmented methods incorporate coding as a tool to assist LLMs in handling graph problems, avoiding the "*repetitive iterative and backtracking operations*" problem. However, existing works rely on graph algorithm APIs and external knowledge bases. On the one hand, they struggle to solve graph problems not covered by APIs and knowledge bases; on the other hand, learning to use APIs rather than the graph algorithm itself does not enhance the problem-solving capabilities of LLMs.

## 4 GRAPH REASONING BENCHMARK

### 4.1 ISSUES WITH CURRENT BENCHMARKS

Previous works on graph reasoning have focused on evaluating fixed algorithm tasks, which may not effectively assess graph reasoning capabilities. Zhang et al. (2024b) demonstrates that training and evaluating LLMs on synthetic classical algorithm datasets primarily teaches the model to learn fixed patterns specific to the seen tasks, without significantly enhancing its general graph reasoning ability. When faced with unseen reasoning tasks or variations in task descriptions, the models fail to generalize effectively. Moreover, there are issues such as the overlap between training and testing tasks, and pre-trained LLMs have knowledge of classic graph algorithms, etc. These factors render current testing tasks overly simplistic and lacking in evaluative significance, which can be easily solved using the simple methods introduced in Section 5. Therefore, it is essential to construct more challenging graph reasoning benchmarks to thoroughly assess the graph reasoning capabilities of LLMs. Other issues like no unified evaluation and incorrect data are discussed in Appendix E.

### 4.2 GRAPHALGORITHM BENCHMARK

To address the issues in previous benchmarks, we collect more challenging graph algorithm problems from algorithm competition platforms, including AtCoder, Codeforces, CodeChef, and Kattis, to construct the GraphAlgorithm benchmark. On the one hand, these problems involve variations of graph algorithms or require the creation of novel graph algorithms, thereby genuinely evaluating LLMs' understanding of graph structures and graph algorithms. On the other hand, the problem descriptions are often framed in real-world scenarios, making them more representative of graph problems encountered in practical applications.

To align with the definition of graph reasoning problems outlined in Section 2, we write dedicated conversion programs for each problem. We integrate the data from test cases into the problem descriptions in natural language, while copying the output formats from the output descriptions in the problem statements. Examples of the problems are provided in Appendix G. After filtering and processing, the final GraphAlgorithm benchmark contains a total of 239 different graph problems and 3,041 test instances, offering a comprehensive and challenging evaluation suite for assessing the graph problem-solving capabilities of LLMs.

## 5 A SIMPLE REASONING-THEN-CODING METHOD

Inspired by findings in Figure 2 and how graph experts solve graph reasoning problems, we present a simple and effective method called Simple-RTC. Simple-RTC decouples graph reasoning and coding, guiding LLMs to reason first and then code to realize. Simple-RTC consists of four steps: Formatting, Extracting, Reasoning and Coding. Figure 3 illustrates the pipeline of Simple-RTC.

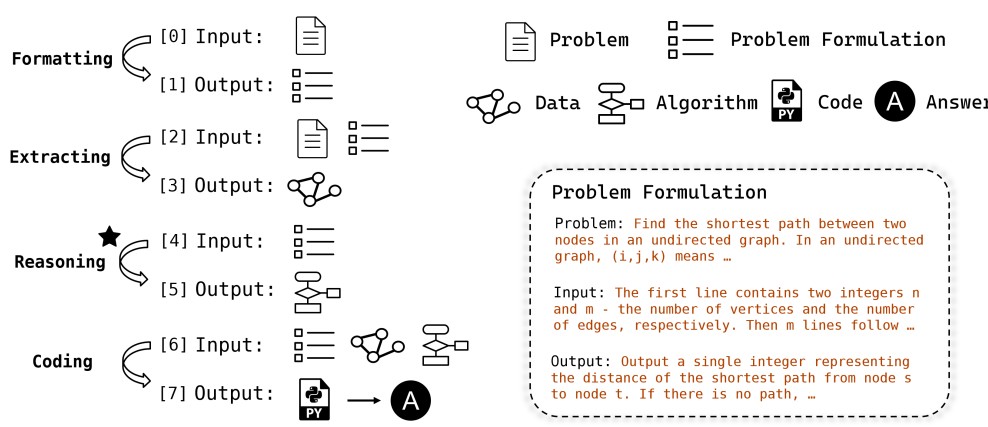

Figure 3: The Pipeline of Simple-RTC.

## 5.1 FORMATTING AND EXTRACTING

The primary reason for the low accuracy of language-based models is that LLMs struggle to precisely memorize complex graph structures, leading to errors in reasoning steps. Additionally, excessive tokens used to describe the graph structure in the context interfere with the LLMs' analysis of the graph problem itself. To address this, we first perform formatting to extract data-free problem descriptions from the problem and provide standardized input-output requirements, as shown in the Problem Formulation in Figure 3. On one hand, this helps the subsequent reasoning to focus on graph problem analysis and graph algorithm design. On the other hand, it helps align the data obtained by extracting, the algorithm generated by reasoning, and the code generated by coding under standardized input-output requirements. We use few-shot prompting to guide the formatting process, with detailed prompts provided in Appendix B.1. After obtaining the Problem Formulation, we instruct LLMs to extract data from the problem using regular expressions based on the standardized input requirements, with detailed prompts provided in Appendix B.2. The problem formulation enables extracting, reasoning, and coding to collaborate effectively, allowing the designed algorithms to be correctly implemented as code according to requirements, and the code to smoothly process the extracted data to obtain the final answer.

## 5.2 REASONING

Reasoning is the key to solving complex graph reasoning problems. However, according to experimental results from Prograph (Li et al., 2024a), LLMs often encounter various code execution errors when generating code to solve graph algorithm problems, leading to a low one-shot success rate across reasoning models. Moreover, strong reasoning models, such as DeepSeek-R1 (DeepSeek-AI, 2025), require significant time to process complex problems, making repeated calls to such models for correct code generation highly costly. Observing that the core results of strong reasoning models in handling graph problems can be stored as algorithm descriptions or pseudocode, we leverage powerful reasoning models as the base model for the reasoning step, specifically tasked with designing algorithms for graph problems. The reasoning step takes the standardized problem formulation from the formatting step as input and focuses on algorithm design without being distracted by graph data. The algorithm descriptions or pseudocode generated by reasoning encapsulate the essential information for solving graph problems, including the core algorithmic logic, key data structures, and step-by-step solution approaches, which can subsequently guide more efficient coding LLMs in generating executable code. Examples are shown in Appendix D.1.

## 5.3 CODING

To improve the accuracy and scalability of solving graph reasoning problems, we choose efficient coding LLMs (e.g., GPT-4o-mini) to generate code to solve graph problems. The coding step takes

the algorithm provided by the reasoning step and the problem formulation provided by the formatting step as input, and generates code to solve the problem. The generated program processes the data obtained from the extracting step and provides the final answer according to the standardized output requirements. Furthermore, since the regular expressions from extracting, the algorithms from reasoning, and the code from coding are all designed based on the Problem Formulation, they can be reused for graph problems of the same type, greatly improving the efficiency of solving graph problems. Detailed prompts are provided in Appendix B.

## 5.4 REUSE MECHANISM

For each problem processed by Simple-RTC, we save the original problem, standardized problem formulation, extracting code, reasoning pseudocode, and coding code. These solution packages are indexed using LlamaIndex (Liu, 2022) for semantic retrieval. For a new problem, we retrieve the most similar problem and sequentially apply its extracting and coding code. If execution fails, we invoke the complete Simple-RTC pipeline and add the new solution to the bank. Since existing benchmarks have fixed problem description formats, Simple-RTC can easily reuse codes, making it both accurate and highly efficient.

## 5.5 COMPARISON WITH RELATED METHODS

While Simple-RTC resembles some code-based methods (Skianis et al., 2024; Cai et al., 2024), these approaches require manually crafted example code or algorithm templates, failing to handle novel graph problems. Simply combining reasoning and coding also proves unreliable, as shown in Table 1. We observe graph structure extraction errors, reasoning models solving problems directly in language without generating algorithms, and code execution failures. In contrast, Simple-RTC demonstrates strong robustness and generalization across multiple benchmarks. This stability stems from the Formatting step, which standardizes the task so that extracting, reasoning, and coding collaborate under a consistent framework to reliably solve diverse graph problems.

## 6 EXPERIMENTS

In this section, we present the performance of previous methods and our Simple-RTC on previous benchmarks and the GraphAlgorithm benchmark we proposed.

**Benchmarks.** For previous benchmarks, we consider NLGraph (Wang et al., 2023), GraphQA (Fatemi et al., 2024), GraphWiz (Chen et al., 2024), GraphArena (Tang et al., 2024) and GraphInstruct (Luo et al., 2024) five benchmarks. For GraphWiz, we rectify the test sets of the problematic Hamilton Path and Subgraph Matching tasks. Relevant details are provided in Appendix E. Benchmark statistics and details can be found at Appendix A.

**Baselines.** For language-based method, we select GPT-4o-mini-2024-07-18, GraphWiz (Chen et al., 2024) and the best-performing models on each benchmark for reporting. For GraphWiz, we select GraphWiz/LLaMA2-13B-RFT with the best performance. For code-augmented methods, we evaluate GraphTeam (Li et al., 2024b) and GCoder (Zhang et al., 2024a). For GCoder, we select GCoder-Llama with the better performance. For GraphTeam and Simple-RTC, we select GPT-4o-mini-mini-2024-07-18 as the base model. We evaluate GraphWiz and GCoder using NVIDIA 2 ×A100 (80GB) GPUs.

### 6.1 RESULTS ON PREVIOUS BENCHMARKS

### 6.1.1 MAIN RESULTS

Table 1 presents the comparison results between our proposed Simple-RTC and other models across five benchmarks. We can see that Simple-RTC achieves near-perfect performance on existing benchmarks, surpassing fine-tuning or carefully designed methods. Compared with the base model GPT-4o-mini, Simple-RTC achieves 48% accuracy gains on average, significantly improving the performance of the base model across various benchmarks. Additionally, we observe that fine-tuned models, such as GraphWiz and GCoder, perform well only on their own training datasets, while

Table 1: Accuracy(%) comparison on different benchmarks.

| Methods | NLGraph | Talk like a Graph | GraphInstruct | GraphWiz | GraphArena | Average Rank |
|---|---|---|---|---|---|---|
| ***Language-based*** | | | | | | |
| Previous SOTA | 79.7 | 90.2 | 48.2 | 54.2 | 75.1 | 3.0 |
| GraphWiz | 17.0 | 30.3 | 19.7 | 54.2 | 2.4 | 5.2 |
| GPT-4o-mini | 41.4 | 61.3 | 59.9 | 51.6 | 35.1 | 4.0 |
| ***Code-augmented*** | | | | | | |
| CoT-to-Code | 63.3 | 72.9 | 67.4 | 81.2 | 63.0 | 3.4 |
| ToT-to-Code | 14.2 | 48.2 | 43.7 | 9.4 | 17.9 | 5.0 |
| GCoder-L | 33.3 | 17.6 | 47.6 | 41.1 | 5.6 | 5.4 |
| GraphTeam | 97.8 | 94.5 | 98.8 | 82.9 | 44.3 | 2.2 |
| ***Ours*** | | | | | | |
| Simple-RTC | **99.3** | **99.9** | **99.9** | **92.7** | **97.5** | **1.0** |

exhibiting significant performance degradation on other benchmarks. Details on the performance of each task in each benchmark can be found in Appendix C.

### 6.1.2 ANALYSIS ON GRAPHARENA BENCHMARK

Table 2: Accuracy(%) comparison on small/large graphs of GraphArena benchmark.

| Method | Polynomial-time Tasks | | | | NP-complete Tasks | | | | | Average |
|---|---|---|---|---|---|---|---|---|---|---|
| | CN | CC | SD | GD | MIS | MVC | MCP | MCS | TSP | |
| ***Language-based*** | | | | | | | | | | |
| Previous SOTA | 100.0/98.8 | 99.6/83.6 | 100.0/98.4 | 95.4/60.0 | 99.4/90.0 | 97.2/74.4 | 95.2/63.4 | 49.6/3.6 | 39.2/3.6 | 86.2/64.0 |
| GraphWiz | 0.0/0.0 | 17.5/7.5 | 2.8/0.3 | 0.8/2.8 | 2.0/0.0 | 2.0/0.5 | 5.5/0.3 | 0.0/0.0 | 0.0/0.0 | 3.4/1.3 |
| GPT-4o-mini | 88.4/77.6 | 82.6/24.6 | 71.2/42.6 | 33.4/1.6 | 42.8/1.4 | 29.4/7.4 | 53.2/6.4 | 48.6/1.2 | 31.8/0.0 | 53.5/16.6 |
| ***Code-augmented*** | | | | | | | | | | |
| CoT-to-Code | 98.0/87.4 | 87.0/45.0 | 88.2/92.6 | 92.4/98.4 | 98.8/83.6 | 22.6/10.0 | 96.4/84.6 | 50.6/5.2 | 5.8/5.4 | 71.2/56.9 |
| ToT-to-Code | 58.8/54.8 | 19.8/8.2 | 23.2/28.8 | 31.6/19.2 | 30.4/15.4 | 2.8/1.4 | 18.8/8.8 | 6.0/0.2 | 1.6/1.6 | 21.4/15.4 |
| GCoder-L | 0.4/0.6 | 29.4/7.8 | 30.8/12.8 | 4.8/2.6 | 0.0/0.0 | 0.0/0.0 | 0.2/8.8 | 1.2/0.6 | 0.0/0.0 | 7.4/3.7 |
| GraphTeam | 98.0/6.0 | 100.0/100.0 | 100.0/56.0 | 100.0/6.0 | 78.0/26.0 | 14.0/2.0 | 100.0/10.0 | 0.0/0.0 | 0.0/0.0 | 65.6/22.9 |
| ***Ours*** | | | | | | | | | | |
| Simple-RTC | **100.0/100.0** | **100.0/100.0** | **99.2/99.4** | **100.0/100.0** | **100.0/98.6** | **100.0/96.6** | **100.0/100.0** | **97.0/68.6** | **100.0/95.2** | **99.6/95.4** |

We specifically present the results on GraphArena in Table 2, which consists of 4 polynomial-time tasks and 5 NP-hard tasks. We can observe that Simple-RTC achieves a significant improvement over the previous models on NP-hard tasks. This is because the reasoning focus of LLMs has shifted to graph algorithm design, enabling the development of more efficient graph algorithms. Thus Simple-RTC is able to solve small-scale NP-hard problems within the given time. Taking the Traveling Salesman Problem (TSP) as an example, the reasoning step designs a more efficient method based on State Compression Dynamic Programming (pseudocode provided in Appendix D.1), thereby effectively solving the TSP problems in GraphArena (with node counts limited to within 20 nodes).

As shown in Figure 4, Simple-RTC demonstrates a significant improvement compared to the base model GPT-4o-mini, merely by enabling GPT-4o-mini to think through and design graph algorithms, which are then implemented

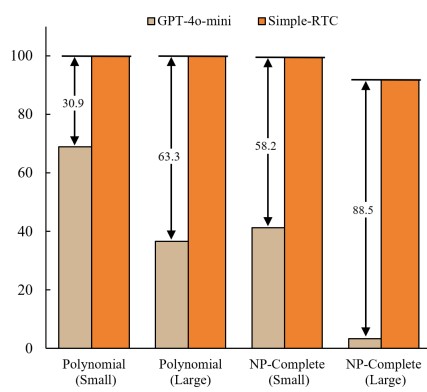

Figure 4: Performance comparison between base model (GPT-4o-mini) and Simple-RTC on GraphArena.

via the coding step. Challenging tasks such as NP-complete problems require algorithms with exponential time complexity to solve, which makes language-based reasoning extremely complex; however, the design of exponential-time algorithms is relatively simple, typically involving brute-force enumeration and backtracking methods. In addition, we can see that the performance improvement on large graphs is more pronounced compared to small graphs. This is because language-based methods are sensitive to graph size due to "*repetitive iterative and backtracking operations*", while Simple-RTC shows strong scalability.

Moreover, GraphArena is a benchmark that is less considered by previous methods. We observe that GraphWiz, GCoder, GraphTeam and prior models perform relatively poorly on this benchmark, which aligns with the issues discussed in Section 3. In contrast, our Simple-RTC demonstrates robustness, achieving strong performance across various benchmarks.

## 6.2 RESULTS ON GRAPHALGORITHM BENCHMARK

### 6.2.1 COMPARISON WITH PREVIOUS METHODS

From the experiments in Section 6.1, we can observe that existing benchmarks do not pose significant challenges for Simple-RTC. Models capable of retrieving external knowledge bases and invoking Graph APIs can easily solve various classic graph algorithm problems, and existing benchmarks fail to evaluate the true graph reasoning ability of these models. Therefore, we evaluate the performance of Simple-RTC and previous methods on the GraphAlgorithm benchmark, with the results presented in Table 3.

Table 3: Performance on GraphAlgorithm Benchmark (%).

| Methods | Average Accuracy |
|---|---|
| *Language-based* | |
| GraphWiz | 0.0 |
| GPT-4o-mini | 14.3 |
| *Code-augmented* | |
| GCoder-L | 3.0 |
| GraphTeam | 8.7 |
| *Ours* | |
| Simple-RTC | **33.7** |

We can see that when faced with unseen and more challenging graph reasoning problems, Simple-RTC still demonstrates reasonable accuracy, outperforming language-based methods and code-augmented methods. In contrast, GraphWiz, which performs well on specific tasks, is almost incapable of solving unseen graph reasoning problems. Additionally, the problems in the GraphAlgorithm benchmark generally lack off-the-shelf graph APIs for direct solutions, requiring extra adjustments or designs. Consequently, GraphTeam and GCoder, which rely on graph APIs and external knowledge bases, are nearly unable to solve problems in the GraphAlgorithm benchmark. Furthermore, the untuned language-based model GPT-4o-mini is capable of solving some graph problems, prompting us to further reflect on the value of fine-tuning LLMs to learn graph reasoning paths or the use of Graph APIs.

### 6.2.2 COMPARISON BETWEEN DIFFERENT REASONING BASE MODEL

Reasoning is the most critical component in addressing graph reasoning problems. To investigate the impact of using different reasoning models as the base model for Reasoning step in Simple-RTC, we keep the base models for Extracting and Coding steps unchanged, and select GPT-4o-mini, o3-mini-high, and Deepseek-r1 as the base models for reasoning step. We test on GraphAlgorithm and the results are shown in Figure 5. We observe that employing a more powerful reasoning model as the base model leads to significant performance improvements, particularly models capable of slow-thinking, such as o3-mini-high and Deepseek-r1. This further highlights that the graph reasoning capability is crucial for solving graph problems. In the future, efforts should focus on enhancing the specialized reasoning abilities of LLMs for graph problems, enabling them to tackle more complex and challenging graph reasoning tasks.

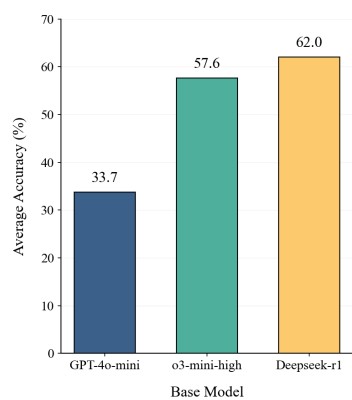

Figure 5: Performance of Simple-RTC with different base model on GraphAlgorithm.

## 6.3 EFFICIENCY ANALYSIS

In terms of efficiency, Simple-RTC has advantages over previous works when handling large-scale graphs and repetitive graph tasks. As the graph size increases, the inference time and cost of previous methods increase rapidly. For example, in an $O(N^2)$ graph algorithm problem, as the number of nodes increases, the number of iterations required grows quadratically, leading to very long inference

lengths. Additionally, long texts further burden the inference of LLMs. In contrast, for Simple-RTC, the reasoning step handles problem formulation, focusing on algorithm design and is almost unaffected by graph size, allowing it to efficiently handle large-scale graph problems. For repetitive graph tasks, such as the fixed graph problems in previous benchmarks, the code generated by the extracting and coding steps can be reused, eliminating the need to repeatedly reason over the same graph task, saving both time and cost. We conducted relevant experiments to validate our analysis, with detailed results provided in Appendix F.

## 7 CONCLUSION

In this work, we identify several issues with existing LLMs for Graph Reasoning methods and Graph Reasoning Benchmarks. We find that base models, e.g., GPT-4o-mini, are largely underestimated due to improper usage. Base models with reasoning focus redirected from replicating graph algorithms to designing them can easily solve most of the graph reasoning tasks in existing benchmarks. To better evaluate the graph reasoning capabilities of LLMs, we construct the GraphAlgorithm dataset. We propose a simple yet strong baseline, Simple-RTC, which can solve graph problems in previous benchmarks and outperforms previous methods on our proposed GraphAlgorithm benchmark. However, we do not conduct dedicated training on the reasoning step in Simple-RTC to enhance its graph reasoning capabilities. Future work will focus on improving specialized reasoning abilities for graph problem solving. The goal of this work is to understand and advance the development of LLMs for Graph Reasoning by facilitating reproducible and robust research. We hope future efforts will focus on improving LLMs' abilities to comprehend graph structures, analyze graph problems, and design graph algorithms.

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

## A    GRAPH REASONING BENCHMARK

Table 4 provides a comprehensive comparison of different graph algorithm benchmarks, including the number of tasks, graph size and test set size.

Table 4: Graph Algorithm Benchmarks Comparison

| Benchmark | Tasks | Number of Tasks | Graph Size | Test Set Size |
|---|---|---|---|---|
| NLGraph | Classic Graph Algorithms | 8 | $\sim 10^1$ | 1,000 |
| Talk like a Graph | Classic Graph Algorithms | 8 | $\sim 10^1$ | 8,000 |
| GraphInstruct | Classic Graph Algorithms | 21 | $\sim 10^1$ | 5,100 |
| GraphWiz | Classic Graph Algorithms | 9 | $\sim 10^2$ | 3,600 |
| GraphArena | Classic Graph Algorithms | 10 | $\sim 10^1$ | 10,000 |
| GraphAlgorithm | Graph Competition Problems | 238 | $\sim 10^3$ | 3,041 |

## B    PROMPT TEMPLATES

### B.1    FORMATTING

Given a problem description, you need to write input and ouput description for this series of problems with different inputs.

Example:

Problem Formulation:

Find the shortest path between two nodes in an undirected graph. In an undirected graph, (i,j,k) means that node i and node j are connected with an undirected edge with weight k. Given a graph and a pair of nodes, you need to output the shortest path between the two nodes. Q: The nodes are numbered from 0 to 9, and the edges are: (0,4,2) (0,8,1) (0,7,7) (0,6,3) (0,3,1) (3,4,4) (3,7,7) (3,8,1) (3,6,10) (4,5,3) (5,6,3) (6,8,1). Give the weight of the shortest path from node 8 to node 5.

Input

The first line contains two integers n and m - the number of vertices and the number of edges, respectively.

Then m lines follow. Each line contains three integers u, v and w, where u and v are the vertices connected by an undirected edge, and w is the weight of the edge.

The last line contains two integers s and t - the source node s and the target node t for which the shortest path distance is to be calculated.

Output

Output a single integer representing the distance of the shortest path from node s to node t. If there is no path, print -1.

Now given a problem desciption:

{}

Only output the input and output description in the following format:

Input

<input_description>

Output

<output_description>

## B.2 EXTRACTING

**Prompts to extract data.**

Problem Formulation:
{}
Write a Python program that use regular expressions to
1. extract input data from the Problem Description
2. convert the input data to standard input strictly following the Input Description below.
Input Description:
{}
Note: 1. The input data in different problems is different, so use regular expressions to extract the input data instead of copying directly.
2. Sometimes the node name may contain spaces. Replace the spaces with _ to prevent reading errors and ambiguity.
3. Sometimes the node name may contain period ".". Pay attention to this when writing the regular expression.
4. Sometimes there may exist example edge like (i,j), remember to remove the example edge. The problem_path and the standard_input_path should be implemented as positional arguments of argparse.
Output the Python program in the following format:
'''python
<python_code_here>
'''

**Prompts to extract problem description and input-output formats.**

Given a problem description, you need to extract the problem itself by substituting data with variables.
Problem Formulation:
{}
Extract Result:
Problem
<extracted_problem>
Input
{}
Output
{}
Now complete the <extracted_problem> part, only output the part in the following format:
Pure Problem
<pure_problem_here>

## B.3 REASONING

Problem Formulation:
{}
Think step by step, design an efficient algorithm to solve this problem, write a corresponding pseudocode.
Reasoning first and summarize your pseudocode in the following format:
Pseudocode
<your pseudocode>

## B.4 CODING

> Problem Formulation:
> {}
> Pseudocode:
> {}
> Write Python code to solve the problem according to the pseudocode.
> Only output your Python code in the following format:
> ```python
> <python_code_here>
> ```

# C ADDITIONAL EXPERIMENTS

Table 5: Performance comparison on NLGraph benchmark in terms of accuracy (%).

| Method | Connect | Cycle | Topo. Sort | Shortest Path | Max. Flow | Bipartite | Hamilton Path | GNN | Overall Average |
|---|---|---|---|---|---|---|---|---|---|
| *Language-based* | | | | | | | | | |
| Previous SOTA | 99.5 | 96.9 | 63.7 | 60.9 | 17.2 | 57.1 | 39.7 | 94.9 | 66.2 |
| Graphwiz | 55.5 | 52.9 | 0.0 | 21.9 | 0.0 | 6.0 | 0.0 | 0.0 | 17.0 |
| GPT-4o-mini | 72.0 | 52.9 | 20.0 | 35.9 | 98.3 | 29.8 | 22.4 | 0.0 | 41.4 |
| *Code-augmented* | | | | | | | | | |
| CoT-to-Code | 82.2 | 23.0 | 63.7 | 34.4 | 72.4 | 51.2 | 98.3 | 87.2 | 64.0 |
| ToT-to-Code | 7.6 | 1.6 | 11.9 | 3.1 | 0.0 | 10.7 | 100.0 | 66.7 | 25.2 |
| GCoder-L | 36.4 | 48.7 | 7.4 | 92.2 | 56.9 | 25.0 | 0.0 | 0.0 | 33.3 |
| GraphTeam | 97.0 | **100.0** | **94.8** | 98.4 | **100.0** | **100.0** | **100.0** | 97.4 | 98.5 |
| *Ours* | | | | | | | | | |
| Simple-RTC | **100.0** | **100.0** | 94.1 | **100.0** | **100.0** | **100.0** | **100.0** | **100.0** | **99.3** |

Table 6: Performance comparison on Talk like a graph benchmark in terms of accuracy (%).

| Method | Node Count | Edge Count | Edge Existence | Node Degree | Connected Nodes | Cycle Check | Shortest Path | Triangle Counting | Overall Average |
|---|---|---|---|---|---|---|---|---|---|
| *Language-based* | | | | | | | | | |
| Previous SOTA | **100.0** | **100.0** | 96.1 | 91.7 | 98.0 | 98.0 | 97.2 | 40.5 | 90.2 |
| Graphwiz | 13.0 | 6.4 | 61.4 | 36.8 | 19.4 | 95.4 | 2.8 | 7.4 | 30.3 |
| GPT-4o-mini | 100.0 | 45.6 | 87.0 | 54.6 | 50.2 | 92.6 | 45.6 | 14.4 | 61.3 |
| *Code-augmented* | | | | | | | | | |
| CoT-to-Code | 77.4 | 98.6 | 37.0 | 98.6 | 80.8 | 98.6 | 45.6 | 46.4 | 72.9 |
| ToT-to-Code | 38.6 | 52.6 | 39.8 | 81.6 | 54.6 | 84.2 | 23.4 | 10.8 | 48.2 |
| GCoder-L | 3.6 | 8.6 | 0.0 | 17.0 | 18.0 | 66.0 | 9.2 | 18.0 | 17.6 |
| GraphTeam | 100.0 | 99.2 | 61.0 | 98.6 | 99.2 | **99.4** | 99.1 | 99.2 | 94.5 |
| *Ours* | | | | | | | | | |
| Simple-RTC | **100.0** | **100.0** | **100.0** | 99.8 | **100.0** | 99.0 | **100.0** | **100.0** | **99.9** |

Table 7: Performance comparison on GraphWiz benchmark in terms of accuracy (%).

| Method | Cycle | Connect | Bipartite | Topology | Shortest | Triangle | Flow | Hamilton | Subgraph | Overall Average |
|---|---|---|---|---|---|---|---|---|---|---|
| *Language-based* | | | | | | | | | | |
| Previous SOTA | 92.8 | 90.3 | 78.5 | 30.3 | 27.5 | 32.8 | 25.8 | **69.5** | 40.5 | 54.2 |
| Graphwiz | 92.8 | 90.3 | 78.5 | 30.3 | 27.5 | 32.8 | 25.8 | **69.5** | 40.5 | 54.2 |
| GPT-4o-mini | 82.0 | 64.8 | 67.3 | 100.0 | 8.5 | 10.3 | 19.3 | 68.0 | 44.3 | 51.6 |
| *Code-augmented* | | | | | | | | | | |
| CoT-to-Code | 99.5 | 99.8 | 85.3 | 97.3 | 94.5 | 88.8 | 95.3 | 32.5 | 37.8 | 81.2 |
| ToT-to-Code | 9.0 | 15.5 | 8.8 | 21.0 | 7.0 | 8.3 | 3.8 | 5.3 | 6.0 | 9.4 |
| GCoder-L | 42.0 | 57.5 | 74.0 | 84.8 | 21.5 | 10.3 | 3.3 | 0.0 | 76.5 | 41.1 |
| GraphTeam | **100.0** | 86.5 | 96.3 | 100.0 | 95.0 | 94.0 | 94.0 | 32.5 | 47.8 | 82.9 |
| *Ours* | | | | | | | | | | |
| Simple-RTC | **100.0** | **100.0** | **100.0** | **100.0** | **100.0** | **100.0** | **100.0** | 42.3 | **91.8** | **92.7** |

To further investigate which specific capabilities Simple-RTC enhances for solving graph reasoning problems, we categorize the problems in the GraphAlgorithm benchmark by their algorithmic types. The benchmark dataset is sourced from four competitive programming platforms: Codeforces, AtCoder, CodeChef, and Kattis. We adopt Codeforces' tagging system to label all problems. Among the 239 problems, 150 from Codeforces already have explicit algorithm tags, while the remaining 89 problems are manually labeled by analyzing their problem statements and solution approaches.

Table 8: Performance comparison on GraphInstruct benchmark in terms of accuracy (%).

| Method | Neighbor | Bipatite | Edge | Connectivity | Degree | DFS | Predecessor | Topological Sort |
|---|---|---|---|---|---|---|---|---|
| *Language-based* | | | | | | | | |
| Graphwiz | 23.7 | 0.0 | 50.7 | 52.3 | 75.7 | 0.0 | 10.0 | 0.0 |
| Previous SOTA | 99.0 | 85.0 | 84.0 | 83.0 | 81.0 | 46.0 | 41.0 | 33.0 |
| GPT-4o-mini | 98.0 | 89.7 | 90.7 | 92.2 | 98.0 | 89.3 | 47.4 | 59.0 |
| *Code-augmented* | | | | | | | | |
| CoT-to-Code | 91.7 | 100.0 | 79.3 | 91.3 | 99.7 | 0.0 | 69.0 | 0.0 |
| ToT-to-Code | 80.0 | 100.0 | 70.3 | 53.7 | 91.7 | 0.0 | 31.7 | 0.0 |
| GCoder-L | 7.3 | 73.7 | 0.0 | 91.7 | 68.5 | 20.0 | 3.7 | 0.0 |
| GraphTeam | **100.0** | 99.0 | 99.3 | **100.0** | **100.0** | 95.7 | 98.0 | 98.3 |
| *Ours* | | | | | | | | |
| Simple-RTC | **100.0** | **100.0** | **100.0** | **100.0** | **100.0** | **100.0** | 99.3 | **100.0** |

| Method | Connected Component | Common Neighbor | Hamiltonian Path | Jaccard | Shortest Path | Diameter | Maximum Flow | Overall Average |
|---|---|---|---|---|---|---|---|---|
| *Language-based* | | | | | | | | |
| Previous SOTA | 83.0 | 23.0 | 18.0 | 14.0 | 14.0 | 13.0 | 6.0 | 48.2 |
| Graphwiz | 10.0 | 23.7 | 0.0 | 0.7 | 32.7 | 2.3 | 14.3 | 19.7 |
| GPT-4o-mini | 39.3 | 49.0 | 33.7 | 17.0 | 57.3 | 23.0 | 15.0 | 59.9 |
| *Code-augmented* | | | | | | | | |
| CoT-to-Code | 28.0 | 99.7 | 0.0 | 99.7 | 97.7 | 96.3 | 58.0 | 67.4 |
| ToT-to-Code | 3.0 | 96.7 | 0.0 | 87.3 | 7.0 | 34.0 | 0.0 | 43.7 |
| GCoder-L | 50.7 | 56.0 | 0.0 | 67.7 | 99.3 | 99.7 | 76.0 | 47.6 |
| GraphTeam | 94.7 | **100.0** | 97.7 | **100.0** | **100.0** | **100.0** | 99.3 | 98.8 |
| *Ours* | | | | | | | | |
| Simple-RTC | **100.0** | **100.0** | **100.0** | **100.0** | **100.0** | **100.0** | **100.0** | **99.9** |

Table 9: Accuracy(%) comparison on GraphArena benchmark. Higher is better.

| Method | Polynomial-time Tasks | | | | NP-complete Tasks | | | | | Average |
|---|---|---|---|---|---|---|---|---|---|---|
| | CN | CC | SD | GD | MIS | MVC | MCP | MCS | TSP | |
| *Language-based* | | | | | | | | | | |
| GraphArena | 72.4/68.6 | 37.1/37.1 | 30.5/16.2 | 29.5/12.4 | 33.3/8.6 | 27.6/19.1 | 35.2/16.2 | 36.2/20.0 | 1.9/0.0 | 33.7/22.0 |
| GraphWiz | 0.0/0.0 | 17.5/7.5 | 2.8/0.3 | 0.8/2.8 | 2.0/0.0 | 2.0/0.5 | 5.5/0.3 | 0.0/0.0 | 0.0/0.0 | 3.4/1.3 |
| GPT-4o-mini | 88.4/77.6 | 82.6/24.6 | 71.2/42.6 | 33.4/1.6 | 42.8/1.4 | 29.4/7.4 | 53.2/6.4 | 48.6/1.2 | 31.8/0.0 | 53.5/16.6 |
| PSEUDO | 54.3/24.8 | 57.1/25.6 | 58.1/34.3 | 33.3/17.1 | 55.2/2.9 | 28.6/13.3 | 40.0/15.2 | 30.5/2.9 | 1.9/0.0 | 39.1/11.3 |
| Claude3-haiku | 76.8/40.6 | 26.0/5.2 | 58.0/35.8 | 11.6/2.0 | 45.0/1.4 | 33.6/7.6 | 48.2/9.0 | 28.2/0.0 | 24.2/0.0 | 39.1/11.3 |
| Deepseek-V3 | 100.0/99.2 | 100.0/93.2 | 98.8/94.2 | 85.0/44.8 | 64.2/30.4 | 36.6/12.0 | 75.4/29.0 | 54.4/1.2 | 37.0/2.0 | 72.4/45.1 |
| Llama-GT | 100.0/98.8 | 99.6/83.6 | 100.0/98.4 | 95.4/60.0 | 99.4/90.0 | 97.2/74.4 | 95.2/63.4 | 49.6/3.6 | 39.2/3.6 | 86.2/64.0 |
| *Code-augmented* | | | | | | | | | | |
| CoT-to-Code | 98.0/87.4 | 87.0/45.0 | 88.2/92.6 | 92.4/98.4 | 98.8/83.6 | 22.6/10.0 | 96.4/84.6 | 50.6/5.2 | 5.8/5.4 | 71.2/56.9 |
| ToT-to-Code | 58.8/54.8 | 19.8/8.2 | 23.2/28.8 | 31.6/19.2 | 30.4/15.4 | 2.8/1.4 | 18.8/8.8 | 6.0/0.2 | 1.6/1.6 | 21.4/15.4 |
| GCoder-L | 0.4/0.6 | 29.4/7.8 | 30.8/12.8 | 4.8/2.6 | 0.0/0.0 | 0.0/0.0 | 0.2/8.8 | 1.2/0.6 | 0.0/0.0 | 7.4/3.7 |
| GraphTeam | 98.0/6.0 | 100.0/100.0 | 100.0/56.0 | 100.0/6.0 | 78.0/26.0 | 14.0/2.0 | 100.0/10.0 | 0.0/0.0 | 0.0/0.0 | 65.6/22.9 |
| *Ours* | | | | | | | | | | |
| Simple-RTC | **100.0/100.0** | **100.0/100.0** | 99.2/99.4 | **100.0/100.0** | 100.0/98.6 | 100.0/96.6 | 100.0/100.0 | 97.0/68.6 | 100.0/95.2 | 99.6/95.4 |

The comparison results are presented in Table 10. Compared to GPT-4o-mini, Simple-RTC achieves significant improvements on graph algorithm problems requiring deeper algorithmic design, such as shortest paths (+41.22%), binary search (+39.79%), data structures (+38.92%), and sortings (+71.92%). This is because the reasoning-then-coding paradigm enables the model to first design algorithmic solutions before implementation, which is particularly effective for problems with clear algorithmic structures. However, Simple-RTC shows weaker performance on mathematical problems like combinatorics (-5.71%), number theory (-11.20%), and dynamic programming (-9.62%), as these problems rely more on mathematical insight and theoretical reasoning rather than algorithmic design.

# D   EXECUTION EXAMPLES

## D.1   PSEUDOCODE FOR TSP

```
Read integer n
Create a list airports and a dictionary name_to_index
for i from 0 to n-1:
    Read airport name and append to airports
    name_to_index[airport name] = i
```

Table 10: Performance comparison on GraphAlgorithm benchmark by algorithm category.

| Tag | Simple-RTC | 4o-mini | Improvement |
|---|---|---|---|
| Sortings | 81.82% | 9.90% | +71.92% |
| Data Structures | 78.32% | 39.39% | +38.92% |
| Brute Force | 63.10% | 41.56% | +21.54% |
| Binary Search | 47.87% | 8.08% | +39.79% |
| Shortest Paths | 46.08% | 4.86% | +41.22% |
| DSU | 50.23% | 39.67% | +10.56% |
| Graphs | 40.19% | 25.31% | +14.87% |
| Bitmasks | 32.54% | 15.94% | +16.59% |
| Constructive Algorithms | 33.38% | 23.50% | +9.88% |
| DFS and Similar | 32.34% | 26.13% | +6.21% |
| Trees | 25.45% | 19.56% | +5.89% |
| Flows | 12.75% | 2.17% | +10.57% |
| Games | 15.38% | 10.00% | +5.38% |
| Geometry | 3.33% | 0.00% | +3.33% |
| Greedy | 24.13% | 25.71% | -1.58% |
| Math | 22.29% | 23.92% | -1.64% |
| DP | 13.98% | 23.60% | -9.62% |
| Combinatorics | 10.00% | 15.71% | -5.71% |
| Number Theory | 14.29% | 25.49% | -11.20% |

```
Read integer m
Initialize a distance matrix dist with size n x n, filled with infinity
for each of m lines:
    Read u_name, v_name, d
    u = name_to_index[u_name]
    v = name_to_index[v_name]
    if d < dist[u][v]:
        dist[u][v] = d
        dist[v][u] = d

// Check if the graph is connected
visited = array of size n initialized to False
queue = new queue
queue.enqueue(0)
visited[0] = True
while queue is not empty:
    current = queue.dequeue()
    for j in 0..n-1:
        if dist[current][j] < infinity and not visited[j]:
            visited[j] = True
            queue.enqueue(j)
if any visited[i] is False:
    print(-1)
    exit

// Initialize dynamic programming table
max_mask = 1 << n
dp = 2D array of size max_mask x n, filled with infinity
dp[1 << 0][0] = 0  // Starting at city 0 with only it visited
```

```
// Process masks in order of increasing number of bits
for bit_count from 1 to n-1:
    for each mask in all masks with bit_count bits and (mask & (1 << 0)) != 0:
        for u in 0..n-1:
            if (mask & (1 << u)) == 0:
                continue
            if dp[mask][u] == infinity:
                continue
            for v in 0..n-1:
                if (mask & (1 << v)) != 0:
                    continue
                if dist[u][v] == infinity:
                    continue
                new_mask = mask | (1 << v)
                if dp[new_mask][v] > dp[mask][u] + dist[u][v]:
                    dp[new_mask][v] = dp[mask][u] + dist[u][v]

// Compute the minimal cycle
full_mask = (1 << n) - 1
result = infinity
for u in 0..n-1:
    if dp[full_mask][u] + dist[u][0] < result:
        result = dp[full_mask][u] + dist[u][0]

if result == infinity:
    print(-1)
else:
    print(result)
```

## E  ISSUES IN PREVIOUS BENCHMARK

**No Unified Evaluation.** Due to the relative ease of constructing classic graph algorithm datasets, most works conduct evaluations on their own datasets. Additionally, the diverse outputs of LLMs and the lack of a unified output format result in significant performance discrepancies for LLMs on the same tasks across different datasets (e.g., shortest path problems that require outputting the path versus those do not). These factors lead to a lack of consistent and fair comparisons among various LLM-based graph reasoning approaches, thereby hindering the development of this field.

**Incorrect Data.** Through meticulous examination, we identify several issues in previous datasets. For instance, the Minimum Spanning Tree task in GraphInstruct lacks edge weight information in its input data; the SubGraph Matching task in GraphWiz was constructed using the incorrect function 'DiGraphMatcher.subgraph_is_monomorphic()', which checks for subgraph monomorphism rather than subgraph isomorphism as described in the task definition. The correct function to use should be 'DiGraphMatcher.subgraph_is_isomorphic()'. Furthermore, errors are also found in PageRank task of GraphInstruct and Hamilton Path problem of GraphWiz. Detailed explanations can be found in the appendix E.1. These flawed task data are subsequently utilized in training and evaluation by later works, which is detrimental to the development of this field.

### E.1  HAMILTON PATH IN GRAPHWIZ

The codes for generate Hamilton Path instances are:

```
def hamiltonian_path(G):
    N = len(G.nodes())
    path = []

    def backtrack(node):
```

```
            path.append(node)
            if len(path) == N:
                return path
            for neighbor in G.neighbors(node):
                if neighbor not in path:
                    result = backtrack(neighbor)
                    if result is not None:
                        return result
            path.pop()
            return "No path found"

        for starting_node in G.nodes():
            path = backtrack(starting_node)
            if path is not None:
                return path

        return "No path found"
```

We can observe that in the backtrack function, all nodes will eventually return 'No path found', and the condition for determining whether to return a path is 'if result is not None'. This will result in the generated Hamiltonian path being incorrect.

We can see a wrong example in the GraphWiz:

> Question:
>
> "Determine whether or not there is a Hamiltonian path in an undirected graph. In an undirected graph, (i,j) means that node i and node j are connected with an undirected edge. Given a graph, you need to output Yes or No, indicating whether there is a Hamiltonian path in the graph. Q: The nodes are numbered from 0 to 4, and the edges are: (0, 1) (0, 3) (0, 4) (0, 2) (1, 3) (3, 4). Is there a Hamiltonian path in this graph?"
>
> Answer in the Dataset: "No"
>
> True answer: Yes. And the counterexample can be (2,0,1,3,4).

### E.2 PAGERANK IN GRAPHINSTRUCT

Similarly, we can see a wrong example in the GraphInstruct:

> Question:
>
> "Given a directed graph: Node <3> is connected to nodes <2>, <1>. Node <2> is connected to nodes <0>, <1>, <4>. Node <1> is connected to nodes <3>, <0>. Node <0> is connected to node <2>. Which node has the largest PageRank value? The dampling factor is 0.85. The number of iterations is 3. The initial PageRank values for all nodes are initialized equally as 1/N, where N is the number of nodes."
>
> Answer in the Dataset: "<1> "
>
> True answer: Node <2>.

## F  EFFICIENCY ANALYSIS

Overall, Simple-RTC's inference time and cost remain at a low level. To further validate our efficiency analysis, we have done the following two experiments:

**Inference time and cost on the NLGraph Benchmark**  We evaluate the inference time and cost of different methods on the NLGraph benchmark, which contains 8 classic graph algorithm tasks. The results are shown in Table 11. Simple-RTC (reused) refers to reusing the previously generated code by Simple-RTC for problems of the same task formulation, which significantly reduces the inference overhead since we only need to run the extractor and execute the code without regenerating the reasoning and implementation.

**Inference time and cost on large-scale graph tasks**  To evaluate how different methods scale with graph size, we conduct experiments on randomly generated shortest path problems with varying node counts (50, 100, and 200 nodes, with 20 problems generated for each size). For each problem, we randomly generate edges between nodes with a density of 0.3 and assign random integer weights between 1 and 100. The average inference time and cost across the 20 problems for each size are reported in Table 12 and Table 13. The results demonstrate that Simple-RTC scales much better with graph size compared to GPT-4o.

Table 11: Performance comparison of different methods.

| Method | Time per problem | Cost per problem |
|---|---|---|
| GPT-4o | 9.92s | 0.0107$ |
| Simple-RTC | 23.89s | 0.0257$ |
| Simple-RTC (reused) | 0.27s | 0.0020$ |

Table 12: Processing Time Comparison by Node Count

| Method | 50 nodes | 100 nodes | 200 nodes |
|---|---|---|---|
| GPT-4o | 142.1s | 157.2s | 207.7s |
| Simple-RTC | 22.4s | 27.0s | 34.6s |

Table 13: Cost Comparison by Node Count

| Method | 50 nodes | 100 nodes | 200 nodes |
|---|---|---|---|
| GPT-4o | $0.169 | $0.175 | $0.187 |
| Simple-RTC | $0.038 | $0.054 | $0.088 |

# G    GRAPHALGORITHM BENCHMARK

**Problem Example.** For more details on the problems and datasets, please refer to the anonymous link.

Assume that you have $k$ one-dimensional segments $s_1, s_2, \ldots s_k$ (each segment is denoted by two integers — its endpoints). Then you can build the following graph on these segments. The graph consists of $k$ vertexes, and there is an edge between the $i$-th and the $j$-th vertexes ($i \neq j$) if and only if the segments $s_i$ and $s_j$ intersect (there exists at least one point that belongs to both of them).

A tree of size $m$ is good if it is possible to choose $m$ one-dimensional segments so that the graph built on these segments coincides with this tree.

You are given a tree, you have to find its good subtree with maximum possible size. Recall that a subtree is a connected subgraph of a tree.

Note that you have to answer $q$ independent queries.

—Input—

The first line contains one integer $q$ ($1 \leq q \leq 15 \cdot 10^4$) — the number of the queries.

The first line of each query contains one integer $n$ ($2 \leq n \leq 3 \cdot 10^5$) — the number of vertices in the tree.

Each of the next $n - 1$ lines contains two integers $x$ and $y$ ($1 \leq x, y \leq n$) denoting an edge between vertices $x$ and $y$. It is guaranteed that the given graph is a tree.

It is guaranteed that the sum of all $n$ does not exceed $3 \cdot 10^5$.

—Output—

For each query print one integer — the maximum size of a good subtree of the given tree.

### Input:
1
10
1 2
1 3
1 4
2 5
2 6
3 7
3 8
4 9
4 10

