# OpenReview forum: "Rethinking and Benchmarking Large Language Models for Graph Reasoning"
_ICLR.cc/2026/Conference — Submitted to ICLR 2026_

### Official Review · Reviewer_182i · 2025-10-17

**Soundness:** 3
**Presentation:** 3
**Contribution:** 3
**Rating:** 6
**Confidence:** 2

**Summary:**

The paper revisits how large language models perform on graph reasoning tasks and argues that existing methods and benchmarks underestimate their potential. It identifies key issues—language-based methods struggle with repetitive reasoning paths, while code-augmented methods overly rely on APIs rather than true reasoning. To address these problems, the authors introduce a new benchmark, GraphAlgorithm, comprising real-world graph competition problems, and propose a simple yet effective baseline Simple-RTC. By decoupling reasoning and coding—first designing algorithms, then implementing them—Simple-RTC achieves near-perfect accuracy on prior benchmarks and significantly outperforms other methods on the new one. The results show that shifting reasoning focus from “replicating algorithms” to “designing algorithms” enables strong generalization across graph tasks.

**Strengths:**

S1. The new GraphAlgorithm benchmark sourced from competitive programming problems is comprehensive, diverse, and realistic, significantly advancing evaluation rigor and generalization testing for graph reasoning.

S2. The proposed Simple-RTC method is elegant and empirically strong, achieving large performance gains across multiple benchmarks and especially on unseen, challenging graph tasks.

**Weaknesses:**

W1. Although the new benchmark is large and diverse, it lacks a fine-grained categorization of graph problems (e.g., traversal, optimization, dynamic programming), which limits more detailed analysis of where reasoning improvements occur.

W2. The paper briefly mentions a “reuse” mechanism in Simple-RTC, but does not specify how such reuse is implemented, detected, or evaluated.

**Questions:**

Q1. How do you ensure that problems in the GraphAlgorithm benchmark are unseen during training of the used LLMs, considering that many competition problems are publicly available online?

Q2. Could you introduce a more fine-grained categorization of graph problems to analyze which categories benefit most from Simple-RTC, and provide experimental results for each category to better reveal the strengths and weaknesses of the approach?

Q3. The paper mentions that Simple-RTC can “reuse previously generated code” for repeated graph tasks—could you elaborate on how this reuse is implemented?

---

> ### Author Response · Authors · 2025-11-20
> **Response to Reviewer 182i**
>
> Thank you for these helpful comments. Our detailed answers are provided below.
>
> > W1&Q2.  Although the new benchmark is large and diverse, it lacks a fine-grained categorization of graph problems, which limits more detailed analysis of where reasoning improvements occur.
>
> We have tagged the graph problems in the benchmark. The updated graphalgorithms.json can be found in the anonymous repository at total_benchmark/GraphAlgorithm. Specifically, the benchmark dataset is sourced from Codeforces, AtCoder, CodeChef, and Kattis. Among these, 150 problems from Codeforces have explicit tags. We manually analyzed and labeled the remaining 89 problems following Codeforces' tagging system.  The comparison results between Simple-RTC and GPT-4o-mini are shown below:
>
> | Tag                         | Simple-RTC Test Pass Rate | 4o-mini Test Pass Rate | Improvement |
> | --------------------------- | ------------------------- | ---------------------- | ----------- |
> | **sortings**                | 81.82%                    | 9.90%                  | +71.92%     |
> | **data structures**         | 78.32%                    | 39.39%                 | +38.92%     |
> | **brute force**             | 63.10%                    | 41.56%                 | +21.54%     |
> | **binary search**           | 47.87%                    | 8.08%                  | +39.79%     |
> | **shortest paths**          | 46.08%                    | 4.86%                  | +41.22%     |
> | **dsu**                     | 50.23%                    | 39.67%                 | +10.56%     |
> | **graphs**                  | 40.19%                    | 25.31%                 | +14.87%     |
> | **bitmasks**                | 32.54%                    | 15.94%                 | +16.59%     |
> | **constructive algorithms** | 33.38%                    | 23.50%                 | +9.88%      |
> | **dfs and similar**         | 32.34%                    | 26.13%                 | +6.21%      |
> | **trees**                   | 25.45%                    | 19.56%                 | +5.89%      |
> | **flows**                   | 12.75%                    | 2.17%                  | +10.57%     |
> | **games**                   | 15.38%                    | 10.00%                 | +5.38%      |
> | **geometry**                | 3.33%                     | 0.00%                  | +3.33%      |
> | **greedy**                  | 24.13%                    | 25.71%                 | -1.58%      |
> | **math**                    | 22.29%                    | 23.92%                 | -1.64%      |
> | **dp**                      | 13.98%                    | 23.60%                 | -9.62%      |
> | **combinatorics**           | 10.00%                    | 15.71%                 | -5.71%      |
> | **number theory**           | 14.29%                    | 25.49%                 | -11.20%     |
>
> We observe that compared to the baseline GPT-4o-mini, Simple-RTC demonstrates significant performance improvement on more complex graph algorithm problems such as shortest paths, union-find, and flow, indicating that Simple-RTC performs better on graph problems requiring deeper algorithmic design. However, Simple-RTC performs poorly on mathematical problems such as combinatorics, number theory, and geometry, as these problems require mathematical insight and theoretical reasoning rather than algorithmic design.
>
> > W2&Q3.  Details of“reuse”mechanism.
>
> For each problem processed by Simple-RTC, we save the original problem, standardized problem description, "extracting" code, "reasoning" pseudocode, and "coding" code. For a new problem, we use LlamaIndex to retrieve the most similar problem and sequentially apply its "extracting" and "coding" code. If execution fails, we invoke the complete Simple-RTC pipeline to process the problem and add it to the problem bank. Since existing benchmarks have fixed problem description formats, Simple-RTC can easily "reuse" codes.
>
>
> > Q1.  How do you ensure that problems in the GraphAlgorithm benchmark are unseen during training of the used LLMs, considering that many competition problems are publicly available online?
>
>
>  We acknowledge that we cannot guarantee that GraphAlgorithm problems are unseen during LLM training, similar to benchmarks like GSM8K, MATH, and HumanEval. However, using the same base model, Simple-RTC significantly improves performance from 14.3% to 33.7% over GPT-4o-mini, demonstrating that our contribution lies in better eliciting graph reasoning capabilities rather than memorization.

---

### Official Review · Reviewer_qvy5 · 2025-11-01

**Soundness:** 1
**Presentation:** 2
**Contribution:** 2
**Rating:** 2
**Confidence:** 4

**Summary:**

This paper critiques existing LLM-based graph reasoning methods and benchmarks, proposing a new benchmark (GraphAlgorithm) and a baseline method (Simple-RTC). Simple-RTC decouples algorithm design (via LLM reasoning) from implementation (via LLM coding), showing strong performance. However, its core idea—separating reasoning and coding—is a standard CS practice, which is not novel.

**Strengths:**

1) The paper provides a thorough and largely justified critique of the current landscape, highlighting real limitations in both existing methods and benchmarks.

2) The Simple-RTC pipeline, while arguably simple, demonstrates strong performance across a wide range of tasks, effectively serving as a  state-of-the-art baseline.

**Weaknesses:**

1) Limited Novelty in Method: The primary weakness is the lack of novelty in the proposed Simple-RTC method. It is essentially a smart prompting strategy that leverages the emergent capabilities of very large, general-purpose LLMs. The "method" itself has minimal technical depth. The four-step pipeline (Formatting, Extracting, Reasoning, Coding) is a sensible engineering workflow, but it is not a scientific innovation on par with new model architectures or training techniques.

2) Overstated Claims: The paper's language, such as "re-thinking the direction," overstates its conceptual contribution. The idea of designing algorithms before implementing them is normal practice and not new.

3) Scalability and Cost: The paper briefly mentions efficiency but does not adequately address the significant inference cost and latency introduced by sequentially querying two large LLMs (one of which is a slow "reasoning" model like DeepSeek-R1). This limits the practical applicability of the proposed approach.

4) Benchmark Baseline Context: As mentioned, the performance on the new GraphAlgorithm benchmark lacks context. Without a non-LLM baseline, it's hard to gauge the absolute achievement of 35.1% accuracy.

**Questions:**

1) Beyond the pipelining of two off-the-shelf LLMs, what is the specific novel technical contribution of the Simple-RTC method itself? How does it differ fundamentally from the established software engineering practice of writing pseudocode before code?

2) The performance seems heavily dependent on the choice of the external "reasoning" LLM (e.g., DeepSeek-R1). To what extent are the reported gains attributable to the Simple-RTC framework versus simply using a more powerful, general-reasoning base model? Could a version of GPT-4o-mini prompted to "think step-by-step" and then code in a single step be a competitive baseline?

3) What is the total API cost and average latency per problem for Simple-RTC on the GraphAlgorithm benchmark? How does this compare to a single-call baseline, making the method's practical feasibility clear?

4) Does the "Formatting" step require manual, problem-type-specific prompt engineering for new tasks, or is it truly generalizable? This impacts the method's scalability to unseen problem types.

---

> ### Author Response · Authors · 2025-11-20
> **Response to Reviewer qvy5**
>
> Thank you for these helpful comments. Our detailed answers are provided below.
>
> > W1&W2&Q1. Limited Novelty in Method. Overstated Claims.
>
>
> We would like to emphasize that our focus lies in "rethinking" the current approach to graph reasoning. Our key contribution is demonstrating that a carefully designed and robustly executable pipeline can surpass the existing model architectures and models fine-tuned on large amounts of data, while nearly solving existing graph reasoning benchmarks. To achieve this, we meticulously designed the workflow and prompts to standardize problem formats, separate problem descriptions from data, and guide models to organize algorithms in pseudocode. These designs enable the model's reasoning to focus specifically on algorithm design while enhancing the overall robustness of the pipeline. We note that previous graph reasoning methods have indeed focused primarily on algorithm reproduction. This is precisely why we use "rethinking"—to highlight the need for redirecting current research efforts toward graph algorithm design.
>
> > W3&Q3. Scalability and Cost
>
> Solving complex problems requires substantial reasoning time. By separating the problem from the data, our approach ensures that reasoning time is nearly unaffected by graph scale. For efficiency considerations, lighter models such as GPT-4o-mini can be employed, which can nearly solve existing graph reasoning benchmarks as demonstrated in our experiments. For the GraphAlgorithm benchmark, the average cost per problem is 0.00117 USD with an execution time of 32.8s; for single-call, the average cost is 0.0004 USD with an execution time of 14.2s.
>
> > W4. Benchmark Baseline Context
>
> To the best of our knowledge, there are no non-LLM methods capable of solving diverse graph algorithm problems. Moreover, the pass rates in these problems do not accurately reflect human-level performance due to repeated submissions. The base model GPT-4o-mini can serve as a baseline, and Simple-RTC improves upon it from 14.3% to 33.7%.
>
>
> > Q2.  To what extent are the reported gains attributable to the Simple-RTC framework versus simply using a more powerful, general-reasoning base model? Could a version of GPT-4o-mini prompted to "think step-by-step" and then code in a single step be a competitive baseline?
>
>  As shown in main Table1, by guiding LLMs to focus its reasoning on the design of graph algorithms, Simple-RTC enhances the performance of the base model GPT-4o-mini by 39% to 62% across different benchmarks. We test GPT-4o-mini prompted to "think step-by-step" and then code in a single step on 5 benchmarks. We find that it suffers from data extraction errors, attempts to solve problems through language reasoning, and exhibits a high probability of code execution errors on complex graph problems. The results are shown below:
>
> | Benchmark | CoT-to-Code | Simple-RTC |
> | --- | --- | --- |
> | GraphWiz | 81.1 | 92.7 |
> | NLGraph | 63.3 | 99.3 |
> | GraphInstruct | 67.3 | 99.9 |
> | TalkLikeAGraph | 72.8 | 99.9 |
> | GraphArena | 63.0 | 97.5 |
>
>
> > Q4. Does the "Formatting" step require manual, problem-type-specific prompt engineering for new tasks, or is it truly generalizable?
>
>
> No，we do not employ task-specific prompts. Instead, a uniform prompt is applied consistently across all tasks. The prompt used in 'formatting' is as follows:
>
> ```python
> Given a problem description, you need to write input and ouput description for this series of problems with different inputs.
> Example:
> Problem Description:
> Find the shortest path between two nodes in an undirected graph. In an undirected graph, (i,j,k) means that node i and node j are connected with an undirected edge with weight k. Given a graph and a pair of nodes, you need to output the shortest path between the two nodes. Q: The nodes are numbered from 0 to 9, and the edges are: (0,4,2) (0,8,1) (0,7,7) (0,6,3) (0,3,1) (3,4,4) (3,7,7) (3,8,1) (3,6,10) (4,5,3) (5,6,3) (6,8,1). Give the weight of the shortest path from node 8 to node 5.
>
> ### Input
> The first line contains two integers n and m - the number of vertices and the number of edges, respectively.
>
> Then m lines follow. Each line contains three integers u, v and w,  where u and v are the vertices connected by an undirected edge, and w is the weight of the edge.
>
> The last line contains two integers s and t - the source node s and the target node t for which the shortest path distance is to be calculated.
>
> ### Output
> Output a single integer representing the distance of the shortest path from node s to node t. If there is no path, print -1.
>
> Now given a problem desciption:
> {}
>
> Only output the input and output description in the following format:
> ### Input
> <input_description>
>
> ### Output
> <output_description>
> ```
>
> The example in the prompt is the shortest path problem, but Simple-RTC achieves strong performance on 56 tasks across 5 benchmarks, demonstrating robust reproducibility and generalizability.

---

### Official Review · Reviewer_VcZR · 2025-11-01

**Soundness:** 4
**Presentation:** 4
**Contribution:** 4
**Rating:** 6
**Confidence:** 4

**Summary:**

Authors argue that current LLM graph-reasoning benchmarks mostly test pattern-matching rather than real reasoning.
Paper introduces GraphAlgorithm, a harder benchmark of 239 algorithmic graph problems, and a Simple-Reasoning-Then-Coding pipeline that first designs and then codes algorithms.

**Strengths:**

- Introduces GraphAlgorithm with 239 problems, a more realistic and harder than prior pattern-heavy sets.

- The Simple-RTC pipeline (reason -> code) is conceptually clean, easy to reproduce, and decouples thinking from execution.

- Shows consistent gains over base GPT-4o-mini and prior methods

- Works to explain why language-only methods struggle (iterative/backtracking) and why naive code-augmented setups may sidestep real reasoning.

**Weaknesses:**

- Benchmark is contest-style algorithmic problems; it’s unclear how insights can be transferred to structural tasks (molecular graphs, KGs, heterogeneous graphs).

- Many contest problems involve modest graph sizes; the paper doesn’t convincingly demonstrate robustness to very large graphs or long inputs.

- We see accuracy gains but ablations on representation choices (node index shuffling, graph description formats) and cross-domain transfer are limited.

**Questions:**

- What breaks first as graph size grows -> reasoning token budget, code generation reliability, or runtime? Any results with long-context inputs or compressed graph encodings?

- How sensitive is performance to node renaming/shuffling, re-ordering of edges, or alternative textual representations of the same graph?

---

> ### Author Response · Authors · 2025-11-20
> **Response to Reviewer VcZR**
>
> Thank you for these helpful comments. Our detailed answers are provided below.
>
> > W1. Benchmark is contest-style algorithmic problems.
>
> We would like to clarify that this paper primarily discusses and compares methods and datasets related to graph computation, which differ significantly from those for knowledge graphs (KGs) and causal graphs.  We focus on graph algorithm problems, which are designed to examine models' understanding of both local and global graph structures. These capabilities are fundamental to solving complex graph problems.
>
> > W2&Q1. Large Graphs or Long Inputs. & What breaks first as graph size grows.
>
>
> On one hand, the GraphAlgorithm includes problems with large graphs (the maximum number of nodes is 1149). On the other hand, it is capable of solving graph problems where the graph structure is provided in a file, allowing it to handle even larger graphs. An example is shown below:
> ```python
> Problem:
>
> There is an undirected graph stored in a txt file, each line of the txt file contains the two vertices of an edge, separated by a space. For example, 0 1. It means that node 0 and node 1 are connected with an undirected edge. Q: What is the length of the shortest path from node 9 to node 5?\nA:
>
> Formatting:
>
> Problem Description:
>
> Given an undirected graph ... (Omitted)
>
> Input
>
> The input is provided in a file named input.txt. Each line of the file contains two integers u and v, separated by a space, representing an undirected edge between vertices u and v.
>
> The last line of the input specifies two integers s and t, separated by a space, representing the source node s and the target node t for which the shortest path length is to be calculated.
>
> Output
>
> Output a single integer representing the length of the shortest path from node s to node t. If there is no path, print -1.
> ```
> Note that 'formatting' specifies in the 'input' that the graph will be provided in input.txt, and then 'coding' will read the file during implementation. The relevant code generated is as follows:
> ```python
> /# Read the graph from the txt file
>
> with open('input.txt', 'r') as f:
>
> edges = [tuple(map(int, line.strip().split())) for line in f.readlines()]
> ```
> Therefore, Simple-RTC can handle large-scale graph problems. Due to Simple-RTC's separation of graph problems and data, the reasoning process is almost unaffected by graph scale. If the problem uses file input, performance remains stable until it exceeds the processing capacity of the computer; if using text input, LLMs encounter context copying errors when the context is long.
>
>
> > W3&Q2.  ablations on representation choices (node index shuffling, graph description formats) and cross-domain transfer.>
>
> Our method demonstrates robustness to graph structure and graph description variations. We conducted the following experiments on the NLGraph dataset: 1. Node index shuffling. 2. Graph description changing.
>
> For node indices, we synchronously modify the nodes in the problem statements so that the problem answers remain unchanged. For graph description styles, we adapted each problem accordingly—for example, in Bipartite, "Applicant 3 is interested in job 2" was changed to "Student 3 is interested in project 2"; in Topology, "node 9 should be visited before node 0" was changed to "Course 9 should be taken before Course 0."
>
> The experimental results are shown below:
>
> | Perturb | Connect | Cycle | Topology | Shortest Path | Max Flow | Bipartite | Hamilton | GNN |
> | --- | --- | --- | --- | --- | --- | --- | --- | --- |
> | Simple-RTC | 100.0 | 100.0 | 94.1 | 100.0 | 100.0 | 100.0 | 100.0 | 100.0 |
> | Simple-RTC (index shuffling) | 100.0 | 100.0 | 94.1 | 100.0 | 100.0 | 100.0 | 100.0 | 100.0 |
> | Simple-RTC (formats changing) | 100.0 | 100.0 | 92.8 | 100.0 | 100.0 | 100.0 | 100.0 | 100.0 |
>
>
> As demonstrated, Simple-RTC exhibits strong robustness to graph description variations. Simple-RTC separates problems from data and is therefore unaffected by node index shuffling; for graph description variations, Simple-RTC only requires updating the regular expressions in the 'extracting' stage.

---

### Official Review · Reviewer_Hshs · 2025-11-01

**Soundness:** 3
**Presentation:** 3
**Contribution:** 2
**Rating:** 4
**Confidence:** 4

**Summary:**

This study argues that LLMs are underestimated due to misplaced reasoning focus. By redirecting reasoning toward designing rather than copying algorithms, base models like GPT-4o-mini can already solve most existing benchmarks. To provide a more rigorous evaluation, the authors introduce GraphAlgorithm, a benchmark of 239 graph problems and 3,041 test instances from competition platforms. They also propose a strong baseline, Simple-Reasoning-Then-Coding (Simple-RTC), which first guides LLMs to design algorithms and then implement them. Simple-RTC achieves near-perfect results on existing benchmarks and significantly outperforms prior methods on GraphAlgorithm, paving the way for future progress in LLM-based graph reasoning.

**Strengths:**

- Simple prompting strategy can outperform prior complex methods highlights the importance of task formulation over model architecture.
- Achieving near-perfect accuracy on old benchmarks and clear improvements on the new one suggests the claims are well-supported experimentally.

**Weaknesses:**

- Structural or semantic graph reasoning may be underrepresented (e.g., subgraph matching, reasoning over KG triples, causal graphs). The benchmark may therefore not generalize to real-world graph reasoning tasks beyond algorithmic problem solving.
- The Simple-RTC success may rely heavily on carefully crafted prompts rather than inherent reasoning improvement, raising questions about reproducibility and generality.
- Limited novelty: RTC is conceptually similar to existing Plan-and-Solve or CoT-to-Code paradigms. The paper may need to clarify how Simple-RTC is substantively different.
- Missing related work:
	- CodeGraph: Enhancing Graph Reasoning of LLMs with Code (Cai et al., 2024). This work addresses how LLMs struggle on basic graph algorithm problems when graphs are converted to text descriptions, and proposes encoding the problem as code (i.e., generate a program and execute it) to improve reasoning.
	- Program of Thoughts Prompting: Disentangling Computation from Reasoning for Numerical Reasoning (Chen et al., NeurIPS 2023). Not graph-specific, but key idea: LLMs generate executable code as reasoning steps, improving accuracy on tasks requiring algorithmic structure. One of the base works for “reason → code → execute” pipelines.

**Questions:**

- Can the authors elaborate on the prompt engineering process? I have already seen the prompts in the appendix.

Suggestions
- Appendix C.4 GRAPHINSTRUCT is empty

---

> ### Author Response · Authors · 2025-11-20
> **Response to Reviewer Hshs**
>
> Thank you for these helpful comments. Our detailed answers are provided below.
>
> > W1. Structural or semantic graph reasoning may be underrepresented.
>
> We would like to clarify that this paper primarily discusses and compares methods and datasets related to graph computation, which differ significantly from those for knowledge graphs (KGs) and causal graphs.  We focus on graph algorithm problems, which are designed to examine models' understanding of both local and global graph structures. These capabilities are fundamental to solving complex graph problems.
>
> > W2. The Simple-RTC success may rely heavily on carefully crafted prompts rather than inherent reasoning improvement, raising questions about reproducibility and generality.
>
>
> In practice, we do not employ task-specific prompts. Instead, a uniform prompt is applied consistently across all tasks. For example,  the prompt used in 'formatting' is as follows:
>
> ```python
> Given a problem description, you need to write input and ouput description for this series of problems with different inputs.
> Example:
> Problem Description:
> Find the shortest path between two nodes in an undirected graph. In an undirected graph, (i,j,k) means that node i and node j are connected with an undirected edge with weight k. Given a graph and a pair of nodes, you need to output the shortest path between the two nodes. Q: The nodes are numbered from 0 to 9, and the edges are: (0,4,2) (0,8,1) (0,7,7) (0,6,3) (0,3,1) (3,4,4) (3,7,7) (3,8,1) (3,6,10) (4,5,3) (5,6,3) (6,8,1). Give the weight of the shortest path from node 8 to node 5.
>
> ### Input
> The first line contains two integers n and m - the number of vertices and the number of edges, respectively.
>
> Then m lines follow. Each line contains three integers u, v and w,  where u and v are the vertices connected by an undirected edge, and w is the weight of the edge.
>
> The last line contains two integers s and t - the source node s and the target node t for which the shortest path distance is to be calculated.
>
> ### Output
> Output a single integer representing the distance of the shortest path from node s to node t. If there is no path, print -1.
>
> Now given a problem desciption:
> {}
>
> Only output the input and output description in the following format:
> ### Input
> <input_description>
>
> ### Output
> <output_description>
> ```
>
> The example in the prompt is the shortest path problem, but Simple-RTC achieves strong performance on 56 tasks across 5 benchmarks, demonstrating robust reproducibility and generalizability. This indicates that our approach elicits the inherent graph reasoning capabilities of LLMs.
>
> > W3. Limited novelty: RTC is conceptually similar to existing Plan-and-Solve or CoT-to-Code paradigms.
>
>
> We would like to emphasize that our focus lies in "rethinking" the current approach to graph reasoning. Our key contribution is demonstrating that a carefully designed and robustly executable pipeline can surpass the existing model architectures and models fine-tuned on large amounts of data, while nearly solving existing graph reasoning benchmarks. Compared to Plan-and-Solve and CoT-to-Code, Simple-RTC employs more refined processing to standardize the problem format, separate the problem from the data, and guide the model to organize algorithms in pseudocode. This allows the model's reasoning to focus on algorithm design and enhances the overall  robustness of the process. We test the CoT-to-Code method (based on GPT-4o-mini) across 5 benchmarks and find that it suffers from data extraction errors, attempts to solve problems through language reasoning, and exhibits a high probability of code execution errors on complex graph problems. The results are shown below:
>
> | Benchmark | CoT-to-Code | Simple-RTC |
> | --- | --- | --- |
> | GraphWiz | 81.1 | 92.7 |
> | NLGraph | 63.3 | 99.3 |
> | GraphInstruct | 67.3 | 99.9 |
> | TalkLikeAGraph | 72.8 | 99.9 |
> | GraphArena | 63.0 | 97.5 |
>
>
> > W4. Missing related work.
>
>
>  Thank you for pointing out these works. We will add them to the related work section.  For CodeGraph,  it requires writing specific task descriptions and sample code for each individual problem, which limits its generality. For Program of Thoughts,  it delegates the computational part to code execution, facing similar issues as CoT-to-Code.
>
> > Q1. the prompt engineering process.
>
>
> The prompt is consistent across all tasks. As shown in Fig. 3, Simple-RTC first prompts the LLM to format the standardized problem description and extract data, then provides the standardized problem description to the model for reasoning to generate pseudocode, and subsequently provides the pseudocode to the model for coding. The model then sequentially executes data_extractor.py and code.py to generate the answer.
>
> > Suggestion: Appendix C.4 GRAPHINSTRUCT is empty.
>
>
> The detailed results for GraphInstruct are shown in Table 8. We acknowledge the formatting issue and will remove the subsection in the revision, presenting the tables directly instead.

---

> > ### Comment · Reviewer_Hshs · 2025-11-24
> >
> > I thank the authors for their answer. Nevertheless, a thorough presentation on how the method is different from other similar techniques and comparison with strong baselines is missing. Since this is mostly a prompting approach it should be compared with many others technique to show its effectiveness and difference. My scoring remains.

---

### Author Response · Authors · 2025-12-01
**General Response to all Reviewers**

Dear Reviewers and Area Chair,

We sincerely thank all reviewers for their dedicated time and constructive feedback. We are encouraged that the reviewers recognized the value of our work, specifically highlighting:

- Providing a thorough and justified reflection on the current landscape of LLMs for graph reasoning(specifically graph computation), identifying limitations in both existing methods and benchmarks. [Reviewer VcZR, qvy5]
- Constructing a diverse, realistic, and more challenging graph reasoning benchmark. [Reviewer VcZR, 182i]
- Proposing Simple-RTC, a robust, conceptually clean and easy-to-reproduce method that requires no complex training or architecture, achieving near-perfect accuracy on existing benchmarks and showing clear improvements on our new benchmark. [Reviewer Hshs, VcZR, qvy5, 182i]

We have carefully addressed questions and concerns in our individual responses and have updated the PDF accordingly. **All modifications are highlighted in blue in the revised pdf.** We summarize the key improvements below:

---

**1. Comparison with similar "Reason-to-Code" methods**

We clarify the distinction between Simple-RTC and naive concatenations of reasoning and code approaches in **Section 5.5**. Furthermore, we add the performance of two reason → code methods, "CoT-to-Code" and "ToT-to-Code" in **Table 1**, with detailed results across all benchmarks provided in **Tables 5–9**. The results demonstrate that Simple-RTC exhibits better robustness and generalizability.

**2. Reuse Mechanism**

We describe the reuse mechanism of Simple-RTC in **Section 5.4**, which enables it to efficiently handle repeated graph tasks.

**3. Further Analysis on the GraphAlgorithm Benchmark**

We provide a detailed categorization of each graph problem in the GraphAlgorithm benchmark, analyzing which tasks or capabilities Simple-RTC primarily improves upon over the base model. Results and analysis are provided in **Appendix C**.

---

Regarding the novelty concerns from Reviewers Hshs and qvy5, we would like to emphasize that the focus should not be on whether a work proposes entirely new architectures or conducts large-scale training, but on whether it solves the problem effectively and robustly. Our simple yet robust method demonstrates that proper task formulation and reasoning focus matter more than new architectures or extensive training. We encourage future work on LLMs for graph reasoning to focus on improving graph reasoning capabilities rather than algorithm reproduction abilities.

---

We believe these revisions have substantially strengthened the paper and addressed the concerns raised during the initial review. We have made effort to respond to each point and hope the reviewers will find our responses satisfactory. We remain fully committed to addressing any remaining questions and would be deeply grateful for the opportunity to continue this constructive discussion.

Best regards,

The Authors

---

### Meta-Review · Area_Chair_n5aw · 2026-01-06

**Summary:**

The paper introduces a challenging GraphAlgorithm benchmark with 239 graph problems and 3041 test instances, and proposes a strong baseline, Simple-RTC, which solves graph problems via a four-step pipeline (formatting, extracting, reasoning and coding).

The authors addressed many concerns in the rebuttal, including adding CoT-to-Code baselines, clarifying the prompt design, solving the graph size and representation problems, reporting the time and monetary costs, and providing a fine-grained categorization of graph problems. However, several key concerns remain and continue to affect the overall quality of the paper. In particular, for the GraphAlgorithm benchmark, it is not possible to ensure that the evaluated LLMs have not seen the problems or solutions during training. Moreover, the proposed Simple-RTC method lacks strong novelty, as noted by Reviewer qvy5. It seems like a human-designed workflow within an agentic system for solving graph algorithm problems. Finally, the approach does not extend naturally to semantic graph reasoning, such as reasoning over molecular graphs or knowledge graphs.

The overall recommendation is to reject.

**Reviewer Concerns:**

Reviewer Hshs raised concerns about the lack of support for semantic graph reasoning (e.g., knowledge graphs), potential reliance on prompt design, missing related work implemented Plan-and-Solve or CoT-to-Code paradigms.

Reviewer VcZR also expressed concerns about semantic graph reasoning in molecular graphs and KGs, as well as the size of graphs and different graph description formats.

Reviewer qvy5 questioned the novelty and claim of the proposed method and raised concerns about API cost and prompt dependence.

Reviewer 182i raised the concerns including fine-grained categorization of graph problems, clarification of the code reuse mechanism in Simple-RTC, and assurances that GraphAlgorithm benchmark problems were unseen during LLM training.

The authors addressed many concerns in the rebuttal, including adding CoT-to-Code baselines, clarifying the prompt design, solving the graph size and representation problems, reporting the time and monetary costs, and providing a fine-grained categorization of graph problems. However, several key concerns remain and continue to affect the overall quality of the paper. In particular, for the GraphAlgorithm benchmark, it is not possible to ensure that the evaluated LLMs have not seen the problems or solutions during training. Moreover, the proposed Simple-RTC method lacks strong novelty, as noted by Reviewer qvy5. It seems like a human-designed workflow within an agentic system for solving graph algorithm problems. Finally, the approach does not extend naturally to semantic graph reasoning, such as reasoning over molecular graphs or knowledge graphs.

**Reviewer Scores:**

Reviewer Hshs suggested keeping the current score, while reviewers VcZR, qvy5 and 182i would most likely keep their original scores.

---

### Decision · Program_Chairs · 2026-01-26

Reject